# Foliar Fertilization with Molybdate and Nitrate Up-Regulated Activity of Nitrate Reductase in Lemon Balm Leaves

Kurmet Aitlessov [1], Bibigul Zhumabekova [2], Utemurat Sagyndykov [1], Akmaral Tuyakbayeva [1], Aliya Bitkeyeva [3], Karlygash Zh. Bazarbaeva [1], Abilkhas Mukhtarov [1], Zhadyrassyn Nurbekova [1], Mereke Satkanov [4], Maral Kulatayeva [1], Karlygash M. Aubakirova [1], Ardak Nurysheva [5] and Zerekbay Alikulov [1,*]

1 Department of Biotechnology and Microbiology, L.N. Gumilyov Eurasian National University, Astana 010000, Kazakhstan
2 Higher School of Natural Science, Pavlodar Pedagogical University, Pavlodar 140002, Kazakhstan
3 Department of Biology and Ecology, Toraighyrov University, Pavlodar 140008, Kazakhstan
4 Department of General Biology and Genomics, L.N. Gumilyov Eurasian National University, Astana 010000, Kazakhstan
5 School of Physics and Mathematics, Nazarbayev Intellectual School, Astana 010000, Kazakhstan
* Correspondence: zer-kaz@mail.ru

**Abstract:** The foliar feeding of soil-grown plants with essential elements such as molybdate can restore their Nitrate reductase activity. However, the activity of nitrate reductase under the foliar feeding of plants with molybdate and nitrate in hydroponic conditions has not been investigated. Thus, we wanted to investigate the effect of the foliar feeding of molybdate and nitrate on the nitrate reductase activity in the leaves of lemon balm plants under hydroponic conditions. Nitrate and molybdate solutions were applied by spraying the leaves of lemon balm plants and the nitrate reductase activity was determined by the colorimetric method. The results of our study demonstrated that the application of molybdate and $KNO_3$ solution enhanced the nitrate reductase activity in the leaves of lemon balm plants. Importantly, our results indicate that foliar fertilization with nitrate and molybdate can improve nitrogen metabolism and carbon fixation in the leaves of lemon balm plants under hydroponic conditions.

**Keywords:** Mo-Co; nitrate reductase; molybdenum; foliar fertilization

## 1. Introduction

Metal-containing enzymes and metalloproteins are necessary for the existence and life of all organisms. Metal ions perform various functions in proteins and enzymes. They can stabilize the substrate molecule, the active site of the enzyme, as well as the protein molecule's conformation [1]. In addition, metal ions can be directly involved in catalysis. Some metals can regulate the activity of enzymes. Molybdenum (Mo) and tungsten (W) are the heaviest elements among all transition metals used in living organisms [2]. They are found in a conserved metal center, which is coordinated by one or two pyranopterins in molybdoenzymes (Mo-enzymes) and tungsten-containing enzymes [2]. The trace element Mo is an important component of the active center of several plant Mo-enzymes such as nitrate reductase (NR), xanthine dehydrogenase (XDH), aldehyde oxidase (AO), sulfite oxidase (SO), and mitochondrial Amidoxime Reducing Component (mARC) [3,4]. Meanwhile, W plays an important role in the normal physiology of archaea, which is justified by the fact that archaea have vital tungsten-containing enzymes [2,3].

Various Mo-enzymes perform vital functions for normal plant development and catalyze, almost exclusively, oxidation/reduction reactions [2–15]. NR plays a key role in regulating nitrate assimilation in plants [3,5,6]. With electrons from NADPH/NADH, the enzyme NR catalyzes the reduction of nitrate to nitrite and performs nitric oxide production [3,5,6]. XDH catalyzes the final step of purine oxidation by converting hypoxanthine

or xanthine to uric acid [2,7,8]. AO oxidizes aldehydes and heterocyclic compounds to their corresponding carboxylic acids and catalyzes the oxidation of abscisic aldehyde to abscisic acid [9–11]. SO oxidizes sulfite with molecular oxygen as an electron acceptor, producing sulfate and hydrogen peroxide [13,14]. However, the function of the most recently discovered Mo-enzyme mARC is still being investigated [15]. It is now known that its functions include the reduction of hydroxylated compounds, as well as the reduction of nitrites to nitric oxide [4,15].

The plant Mo-enzymes have a predominantly homo-dimeric structure consisting of two subunits (mARC is an exception) [4–15]. However, plant XDH, AO, NR, and SO have differing domains in their subunits depending on the type of Mo-enzyme [5–15]. In XDH and AO enzymes, each of the two subunits contains two iron-sulfur centers, flavin adenine dinucleotide (FAD), and molybdenum cofactor (Mo-Co) [7–12]. NR contains five conserved domains in each monomer [5,6]. These domains include the Mo-molybdopterin domain, which has a single Mo-Co, a dimer interface domain, a cytochrome b domain, and the NADPH/NADH domain, which combines with the FAD domain to create the cytochrome b reductase fragment [5,6]. SO, as the simplest plant Mo-enzyme, possesses only one redox center in the form of Mo-Co [13,14]. Monomeric mARC, generally found in eukaryotes, has two domains: a MOSC domain, which is found in MOS enzymes involved in Moco sulfurases, and a β-barrel domain [4,15].

In the active center of Mo-enzymes, Mo binds to two thiols of this cofactor [2–4,16–20]. On the other hand, it has been found that the biosynthesis of Mo-Co occurs in the absence of Mo [19]. The Mo atom is incorporated into Mo-Co during the assembly of the Mo-enzyme molecule [18,19]. These cofactors are called pyranopterin and have a tricyclic structure comprising pyrimidine, piperazine, and pyran-dithiolene rings (Figure 1) [2–4,16]. Two sulfhydryl groups (-S-C=C-S-) form a five-membered ene-1,2-dithiolate chelate ring with a Mo atom (Figure 1) [16,17,20]. The Mo atom in the enzyme's active center binds to two adjacent sulfhydryl (-SH) groups of pyranopterin during the catalytic reaction [16,17,20]. However, the Mo-Co structures can vary according to the Mo-enzyme type. In the case of SO, NR, and mARC, Mo-Co directly binds with apoenzymes, but in the case of AO or XOR/XD, and Mo-Co, a final sulfuration undertakes their incorporation into the apoenzymes (Figure 1) [4]. Additionally, there are some MoCo binding proteins which are able to interact with and transfer MoCo [20].

Although Mo is required at a very low concentration for plants, the concentration of Mo needs to be at a high concentration in the aqueous medium for its assimilation by plants under aquaponics conditions [21–23]. Therefore, the foliar feeding of plants is essential to fulfill plants' nitrate nutrition and Mo requirements under aquaponics conditions. Despite numerous studies on mineral nutrient uptake by leaf tissues, many aspects of foliar fertilization are still unknown and need further investigation [24,25]. Nowadays, it is believed that such fertilization of plants is an important support for nutrient application in the nutrient environment [24,25]. Foliar fertilization is most effective when the soil nutrients are deficient [26]. Foliar fertilization is also important in reducing contamination of the aquatic environment with ions of certain elements, such as nitrogen in aquaponics [27]. For plants, nitrate is an easily digestible source of nitrogen [28].

Rana et al. describe that the foliar feeding of plants with Mo in a soil environment with a lack of Mo showed a complete restoration of lost NR activity [23]. This prompted us to hypothesize that the application of Mo and nitrates using a foliar feeding strategy can increase the assimilation of nitrogen compounds and the quality of plant products via the activation of NR. To test this hypothesis, lemon balm (*Melissa officinalis* L.) was chosen as an object. Lemon balm is a well-known medicinal plant from the *Lamiaceae* family [29]. The leaves of this plant are widely used in cooking to add flavor to dishes [29,30]. The plant is also used to treat mental and central nervous system diseases, cardiovascular and respiratory diseases, and various types of cancer, and is also used as a memory enhancer, cardiac tonic, antidepressant, sleep aid, and antidote [30]. The current work presents

research on the effect of the foliar feeding of molybdate and nitrate on the activity of nitrate reductase in the leaves and roots of lemon balm plants grown using hydroponics.

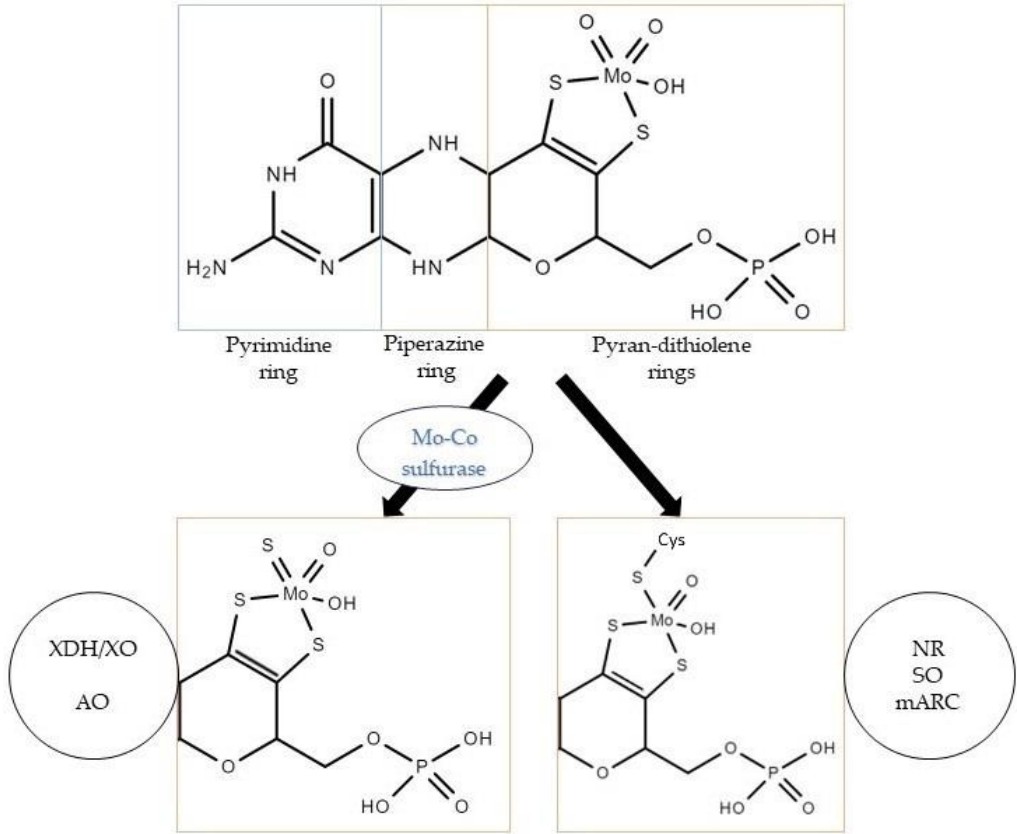

**Figure 1.** Structure of molybdenum cofactor [4]. XDH/XO, xanthine dehydrogenase/xanthine oxidase; AO, aldehyde oxidase; NR, nitrate reductase; SO, sulfite oxidase; mARC, mitochondrial amidoxime reducing component; Cys-cysteine.

## 2. Materials and Methods

### 2.1. Lemon Balm Cultivation

The plant lemon balm was used to investigate the effect of exogenous foliar fertilizers, nitrate and molybdate solution, on the NR activity of leaves under hydroponics conditions. Lemon balm seeds were obtained from the collection of the Kazakh Agrotechnical University S. Seifullina. Seed cultivation was carried out in 3 stages. The first stage was carried out in theory cups on filter paper soaked in distilled water for two days in the dark [31]. At the second stage, lemon balm seedlings were transplanted to soil and cultivated for seven days in white light with a PAR incident photon flux density of 200–230 $\mu$mol m$^{-2}$s$^{-1}$, with a 16-h photoperiod and air temperature of 20–22 °C [32]. In the third stage, 9-day-old seedlings were transplanted into an aquaponic setup and fixed using drainage.

### 2.2. Foliar Fertilization

The plants were grown and the experiment was conducted under hydroponics conditions with 16 h of illumination, day temperature 26 °C, night temperature 20 °C, and relative humidity 60%. Plants were grown for 15 days in a nutrient solution with the complete composition of Hoagland's medium and then transferred for 5 days to a nutrient solution containing no nitrogen sources (without urea, ammonium, or nitrate) [33]. Then, leaves were sprayed with solutions with different concentrations of $KNO_3$ (1.0, 5.0, and 10.0 mM) as a nitrogen source. Tween-20 (0.1%) was added to the solutions as a softening agent for this treatment [34]. The spraying rate was 300 mL per 1 plant. Spraying was carried out using a hand sprayer (hand sprayer with pump and 2-L release valve 64738

PALISAD) [35]. The treatment was carried out between 09:00 and 11:00 in the morning. Plants were sprayed equally with solutions until the leaves were completely wet and the solution dripped off the leaves. During spraying, other plants were covered with plastic bags to prevent contamination of the sprayed solutions. Foliar feeding (leaf spraying) of the plants with a solution of 1.5 µM sodium molybdate ($Na_2MoO_4$) was carried out in the same way and under the same conditions [36].

### 2.3. Determination of the Amount of Nitrate

Nitrate amounts were determined in the leaves every 8 h for 56 h after spraying the leaves. Plant leaves were cut into small pieces ($1.0 \times 1.0$ cm). A total of 100 mg of leaf tissue was boiled for 10 min in 5 mL of distilled water. A total of 0.2 mL of salicylic acid was added to 50 µL of this boiled leaf extract. A total of 4.75 mL of 2 N NaOH was added to this solution after incubation at room temperature for 20 min [37]. The final volume was brought to 5 mL. The absorbance of the resulting colored solution was measured at 548 nm [7,12,38]. The nitrate content was calculated using a calibration curve (µg/g fresh leaves mass).

### 2.4. Determination of NR Activity In Vivo

Leaf pieces (200 mg fresh weight) were soaked for two minutes in 5 mL of a reaction mixture containing 100 mM potassium phosphate buffer (pH 7.5), 25 mM $KNO_3$, and 1% isopropanol [33,39]. Isopropanol increases cell permeability and renders the leaf segments anaerobic when it is in medium. Under anaerobic conditions, the reduction of nitrite to ammonia is inhibited and the conversion of nitrate to nitrite is increased. The reaction mixture was incubated at 30 °C for 30 min in the dark. NR activity was estimated by the amount of formed $NO_2^-$ during the incubation period and the amount released from leaf discs into the medium after boiling for 5 min ($NO_2^-$ per min per mg of protein) [7,12,38]. Protein content was analyzed and equalized by using the Bradford protein assay [40].

### 2.5. Preparation of Samples for Analysis of NR Activity and Determination of Mo-Co Content

Fresh leaves were ground in liquid nitrogen. After evaporation of liquid nitrogen, the frozen powder was mixed with extraction buffer in the ratio of 1 g fresh leaves to 2 mL of 25 mM potassium-sodium-phosphate buffer (PBS) with pH 7.0 containing 10 µM EDTA, 20 µM FAD, 20 µM PMSF, and 10 µM cysteine [41,42]. The supernatant was obtained by centrifugation of the homogenate at $15,000\times g$ for 10 min at 4 °C. For the determination of NR and Mo-Co activity, leaf extracts were passed through Sephadex G-25 (course, Pharmacia Fine Chemicals) in the extraction buffer. Sephadex G-25 was used to separate proteins from low molecular weight compounds that could affect enzyme activity. For analysis, a protein fraction (PF) of fresh leaf extract of lemon balm was used. Protein concentration was measured using the Bradford method [40].

### 2.6. Determination of NR Activity In Vitro

To determine NR activity, 200 µL of supernatant was added to 300 µL of PBS pH 7.0 containing 10 µM EDTA and 20 µM FAD to obtain a final volume of 0.5 mL [33,41]. A total of 50 µL of 2.5 mM electron donors and $KNO_3$ were added to start the reaction. The electron donors that were used were NADH and benzyl viologen (BVH). The mixture was incubated for 15 min at room temperature. Then, 0.5 mL of sulfonamide and 0.5 mL of N-(1-naphthyl)-ethylenediamine were added to the mixture to stain the formed nitrites [7]. The nitrites were stained pink, and the color intensity was measured at a spectrophotometric wavelength of 548 nm [7]. The nitrite content was calculated using a calibration curve ($NO_2-$ per min per mg of protein) [7,12,38].

### 2.7. Determination of Mo-Co Content

One of the reliable methods for the identification of Mo-enzymes is the use of a nit-1 mutant of the fungus *Neurospora crassa* [39,41,42]. In this mutant, the NR apoprotein is

synthesized without a cofactor [42,43]. When Mo-Co isolated from other Mo-enzymes is added to the nit-1 mutant extract, the exogenous cofactor is incorporated into the active center of the NR and restores its activity [44]. All plant molybdoenzyme species contain Mo-Co and the cofactor isolated from them activates the NR of the nit-1 mutant [42,44]. However, when Mo-Co is isolated from Mo-enzymes, molybdenum is immediately separated from the cofactor, and the cofactor does not form a complex with molybdenum outside the apoenzyme molecule [42]. Therefore, when the nit-1 mutant extract is incubated with a cofactor source, a high concentration of exogenous molybdenum is required for the self-assembly of its NR holoenzyme [43].

Cultivation of the nit-1 mutant of the fungus *Neurospora crassa*, obtaining its extract, and determination of the activity of the newly formed nit-1 mutant which is activated by Mo-Co from extracts obtained from leaf discs of plants were carried out according to our previously improved method [41]. To determine the maximum activity of Mo-Co, the source (leaf extract) was heated at 80 °C for 5 min in the presence of glutathione (GSH) and molybdate. The cofactor is bound non-covalently to the apoprotein of molybdoenzyme and, therefore, dissociates from the protein upon thermodenaturation of the enzyme and rapidly loses the molybdenum atom. The cofactor is extremely sensitive to oxygen—its neighboring two SH-groups are oxidized by oxygen and form an irreversible disulfide (-S-S-) bond. The presence of GSH protects them from oxidation. Therefore, the source of Mo-Co can be heated aerobically. When Mo-Co was mixed with the mutant extract, 5 μM was added. It promotes the regeneration of NADPH, which is an electron donor for the newly formed mutant NR. The NR activity of lemon balm leaves, and the newly formed NR mutant nit-1, were expressed in nanomoles of $NO_2^-$ per min per mg of protein. Protein concentration was measured using the Bradford method [40].

### *2.8. Statistical Data Analysis*

For statistical analysis, the GraphPad Prism 8 program was used. The analysis was carried out using two-way ANOVA with Tukey's multiple comparisons, correlation tests, and one-way ANOVA with Tukey's multiple comparisons. At least 5 replicates were used for analysis. Data were accepted as statistically significant at $p < 0.05$.

## 3. Results

### *3.1. Effect of Foliar Fertilization with Potassium Nitrate on Nitrate Absorbtion and NR Activity within 56 h after Spraying*

In plant leaves, potassium nitrate showed a higher inductive effect than $NaNO_3$, probably because of the $K^+$ ion in controlling the nitrate channel or the influence of this ion on nitrate penetration into the leaves and the induction of NR [45]. Therefore, we first determined the optimal concentration of $KNO_3$ and when it can penetrate the internal tissues of lemon balm leaves at the maximum rate (Figure 2). The amounts of absorbed nitrate were then determined in the leaves every 8 h for 56 h. The amount of nitrate was determined directly from the extract itself without gel filtration through Sephadex.

The experimental results presented in Figure 2A demonstrate that maximum nitrate absorption after foliar fertilization with $KNO_3$ 1 mM is achieved after 48 h. This is justified by the fact that the value of absorption of nitrate after 56 h is statistically insignificant, along with the value at 48 h ($p > 0.05$). However, after foliar fertilization with $KNO_3$ 5 mM, the maximum nitrate uptake was achieved after 40 h (Figure 2B), whereas, after spraying the leaves with 10 mM $KNO_3$, the maximum nitrate uptake was achieved after 32 h (Figure 2C). According to a comparative analysis of the dynamics of the uptake of different concentrations of potassium nitrate, the maximum amount of nitrate is absorbed at 40 h. A two-way ANOVA with multiple comparisons showed that there was no statistical significance in the content of nitrate in the leaves after treatment at 40, 48, and 56 h ($p > 0.05$). As a result, it was determined that the maximum amount of nitrate was absorbed by the leaves of lemon balm at 40 h after spraying with $KNO_3$ solution, and the optimum

concentration was 5.0 mM. In contrast, neither in vivo nor in vitro NR activity was detected in the leaves 56 h after spraying with nitrate at the indicated concentrations.

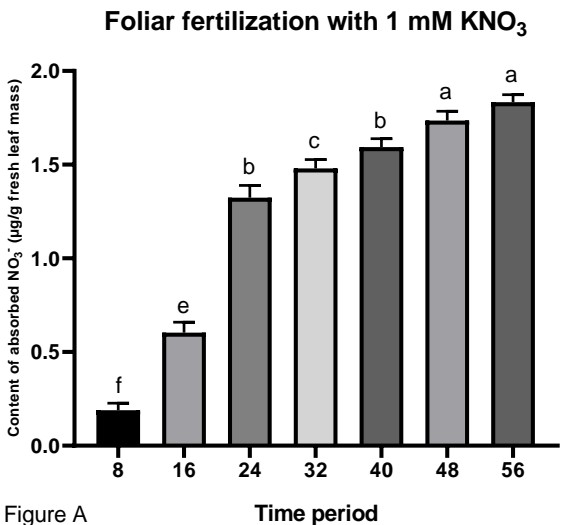

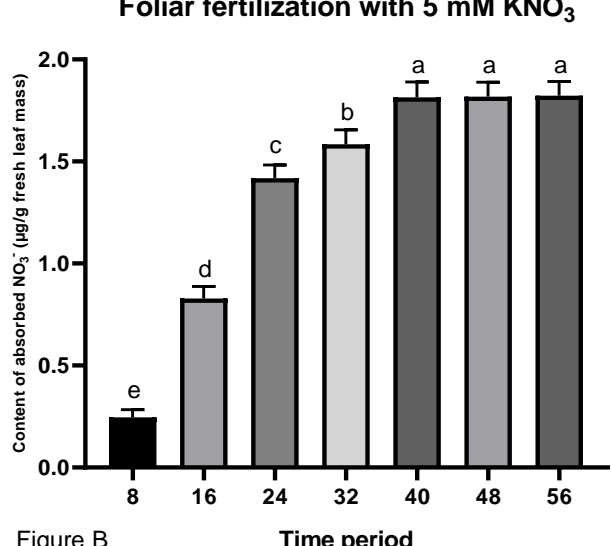

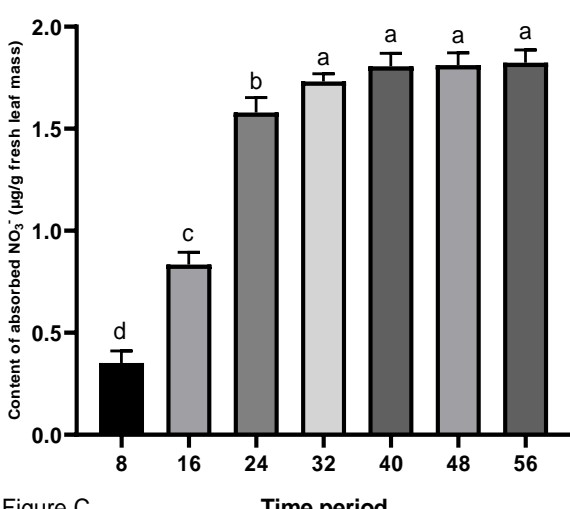

**Figure 2.** Content of absorbed nitrate dependent on the time period after foliar fertilization with KNO$_3$ 1 mM (**A**), 5 mM (**B**), and 10 mM (**C**) by leaf discs of lemon balm plants. In (**A–C**), statistical significance ($p < 0.05$) between groups (time after spray) is shown in different small letters of the alphabet.

### 3.2. Effect of Foliar Fertilization with Sodium Molybdate on NR Activity within 56 h after Spraying

In the following experiments, lemon balm leaves were sprayed with molybdate (Na$_2$MoO$_4$) solution 40 h after nitrate treatment. Analysis of the literature showed that molybdate solutions were used to spray leaves at concentrations ranging from 1.0 to 2.0 μM (physiological concentration of Mo is about 0.01 μM) [36]. Therefore, we used 1.5 μM molybdate to spray lemon balm leaves. Then, in vivo, the NR activity in the leaves was determined every 8 h for 56 h. However, we were unable to detect in vivo and in vitro NR activity within this time frame. As shown in the previous experiment (Figure 1), the maximum level of nitrate was absorbed by the leaves after 40 h. The synthesis of NR molecules in leaves should have been induced after such time. Therefore, it was assumed

that molybdate was absorbed more slowly by the leaves and that at such a time Mo was not accumulated sufficiently to incorporate into the active center of NR.

### 3.3. Mo-Co Content in the Lemon Balm Leaves within 56 h after Potassium Nitrate Spraying

To examine this assumption that the 56 h Mo was not accumulated sufficiently, leaves were homogenized after nitrate spraying, and their extracts were obtained for the detection of Mo-Co content (Figure 3). As previously defined, the Mo-Co domain of NR is stable at 70 °C for up to 5 min [46]. This heat treatment reversibly denatures this enzyme domain and will be available to the active center, thus Mo-Co becomes accessible to exogenous molybdate ions. The presence of the reducing agent, GSH, is also required. The cofactor also becomes available to oxygen during denaturation. Oxygen oxidizes the neighboring -SH groups, reducing them to a disulfide form which cannot participate in the catalytic reaction of the enzyme. Thus, we carried out experiments on the heat treatment of PF leaf extract at 70 °C in the presence of exogenous molybdenum and GSH and, following that, the determination of NR activity in vitro.

**Figure 3.** Mo-Co activity in lemon balm leaves in control leaves and after nitrate spraying. Different letters indicate significant ($p < 0.05$) differences. Small letters indicate differences within groups. Asterisks indicate statistical significance ($p < 0.05$) between groups.

The results showed that nitrate absorbed by the leaves did induce NR syntheses, but it did not contain molybdenum in the active center and was inactive. This was confirmed by using an extract of the nit-1 mutant of *Neurospora crassa* (as mentioned above, the Mo-defective NR mutant is activated by an exogenous cofactor). As shown in Figure 3, from 24 h after nitrate spraying, the Mo-Co activity increases because the cofactor content increases. This is due to new syntheses of Mo-Co in NR molecules. This correlates with the dynamics of accumulation of sprayed nitrate in plant leaves.

In control leaves, where distilled water was sprayed instead of nitrate solution, the detected Mo-Co activity amount was at the same level without change until 56 h. The source of Mo-Co in control leaves (without nitrate) comes from the other Mo-enzymes such as AO, XDH, SO, and mARC [2–15]. According to these results, it is clear that 56 h is insufficient for the uptake of molybdate required for NR activation in lemon balm leaves. Therefore, in the following experiments, molybdate was sprayed every other day for 13 days (to prevent the accumulation of $KNO_3$ in leaf tissues in the absence of its assimilation, a solution of 1.0 mM nitrate concentration was sprayed on the leaves every day). The determination of molybdenum in biological materials is an expensive and time-consuming procedure;

therefore, activation of the NR enzyme is an indicator that molybdate penetrates leaf tissues and is incorporated into the active center of NR.

### 3.4. Effect of Foliar Fertilization with Sodium Molybdate and Nitrate on NR Activity within 15 Days after Spraying

The activity of NR was determined using different electron donors in PF extracts: NADH—physiological donor, and reduced BVH—artificial donor. The results for the in vivo NR activity are demonstrated in Figure 4. As can be seen from Figure 4, statically significant in vivo NR activity was achieved after 6 days of spraying with nitrate and sodium molybdate compared to the first day. However, for the first 8 days, no activity was detected. It was also determined that the in vivo NR activity increases sharply on day 10 and reaches its maximum on day 12. The in vivo NR activity on day 14 showed a similar level to knock 12 ($p > 0.05$).

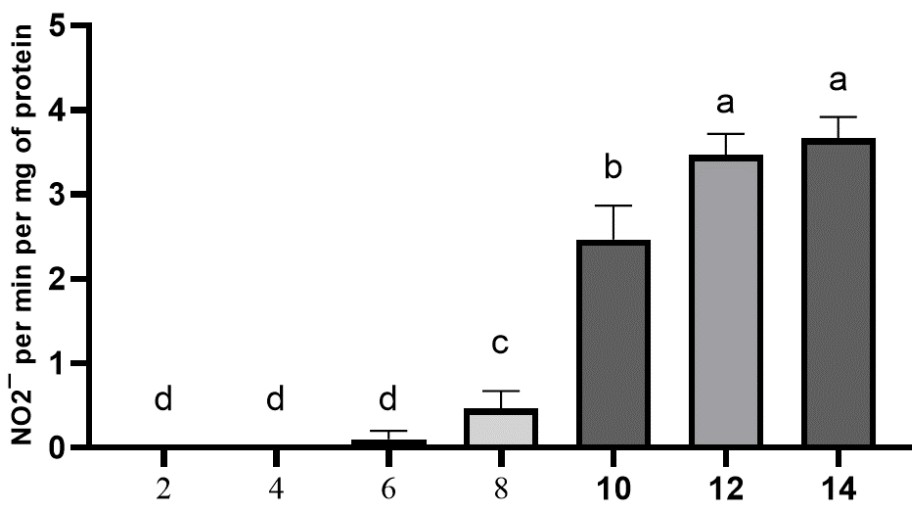

**Figure 4.** The in vivo NR activity of lemon balm leaves sprayed with molybdate and nitrate solution. The significance statistic was $p < 0.0001$ by one-way ANOVA. Different letters indicate significant differences.

The results for the in vitro NR activity are shown in Figure 5. The results of these experiments show that in vitro NR activities started to be detected 8 days after spraying with molybdate solution on the lemon balm leaves. In vitro NR activity was evident with NADH and BVH, and the same pattern was maintained with each electron donor (Figure 5). The in vitro NR activity increases sharply on day 10 and reaches a maximum on day 12. The maximum in vitro activity of the enzyme was shown by using BVH as an artificial electron donor. Moreover, the activity of BVH-NR was significantly higher than that of NADH-NR.

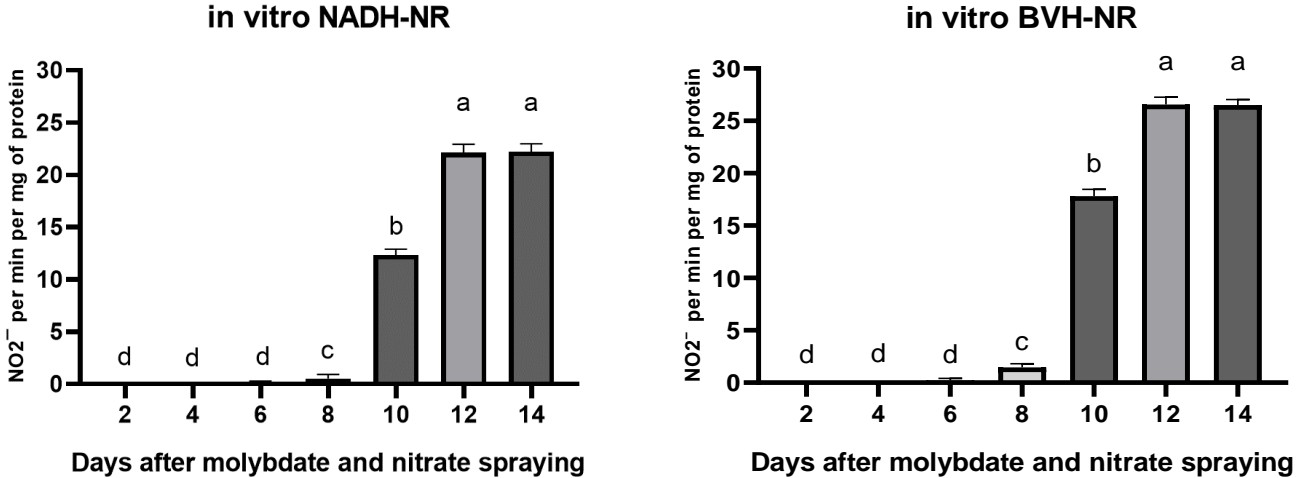

**Figure 5.** The in vitro NR activity of lemon balm leaves sprayed with molybdate and nitrate solution. The significance statistic was $p < 0.0001$ by one-way ANOVA. Each group of NR activity statistical significance is calculated separately. Different letters indicate significant differences.

## 4. Discussion

In this work, the NR activity of leaves was used as a marker to establish the incorporation of exogenous molybdenum into Mo-enzyme molecules. We selected NR to establish the incorporation of exogenous Mo into Mo-enzyme by its activity. Plant NR is an inducible Mo-enzyme that synthesizes in the presence of its substrate, nitrate, within the cell [3,5,46–48]. NR activity occurs only in the presence of Mo in its active center [3–5,16,19]. The other Mo-enzymes are constitutive, they can be synthesized continuously regardless of the presence of their substrates [4,9,10,13–15]. Therefore, it is difficult to establish the exact time of incorporation of exogenous Mo (unlabeled) into their molecules. Thus, by spraying nitrate and molybdate on plant leaves, it is possible to induce and control the timing of NR biosynthesis and its activation by molybdenum.

The plant NR molecule is complex, consisting of an NADH- and FAD-binding diaphorase domain and a Mo-containing domain [43–48]. It should be noted that nitrate reduction occurs in the Mo-Co domain. The diaphorase domain is very sensitive to temperature, so at temperatures above 35 °C it can denaturate within a few minutes [42]. The Mo-containing domain is more stable, and it can retain active conformation until 50 °C for 4–5 min [42,49]. On the other hand, isolating Mo-Co from molybdoenzyme using heat treatment at 80 °C in the presence of molybdate and GSH will lead to the denaturation of the domain completely and will release the cofactor [41–44]. The Mo-Co domain can be partially denatured when gentle heat treatment until 50 °C was used, and this will result in access to the cofactor for exogenous molybdate and GSH [41].

NR activity can be determined using natural (NADH and $FADH_2$) and artificial electron donors [2,3,7,38]. The Mo-Co domain can directly accept electrons from artificial donors such as reduced BVH and transfer them to nitrate, reducing it to form nitrite [41]. Therefore, after heat treatment at 50 °C, NR activity was determined using dithionite (sodium hydrosulfite) as a reducing agent and BVH as an artificial electron transfer to nitrate [44].

Syntheses of NR molecules in the roots of seedlings start 4–8 h after the addition of nitrate and reach a maximum at 12–15 h depending on the plant species [36]. Analysis of the literature shows that the optimum concentration for NR induction in roots depends on the plant species and ranges from 20 to 70 mM in a nutrient medium [44]. The foliar feeding of the plants showed that the concentration of nitrate for NR induction in leaves was significantly lower as compared to root feeding: it was about 5.0 mM [50].

These results suggest that foliar fertilization has an advantage under aquaponics. The foliar nitrate is well-assimilated by the leaves of aquaponics plants and fully fulfills

the required nitrogen source of the plants. On the other hand, it might be a good supplement for root nitrate, which can only form from fish waste through nitrification by microorganisms [50]. NR is the first enzyme of the nitrate nitrogen assimilation pathway in plants [2–5,51]. In higher plants, NR can be rapidly modulated by environmental conditions such as light, $CO_2$, or oxygen availability [52,53]. High aeration inactivates NR in plant roots [54]. The activity of NR in plant roots and the level of nitrate assimilation can be lower in aquaponics water, which is specifically oxygenated.

Mo stimulates nitrogen and carbon metabolisms to increase photosynthesis and plant productivity [36,46,49,55]. Recent studies have shown that Mo also directly affects photosynthesis because of its involvement in chlorophyll biosynthesis and the stability of the photosynthetic apparatus [56]. Consequently, enhanced photosynthesis may also contribute to an increase in the activity of NR [57,58]. Thus, the foliar fertilization of aquaponics plants with nitrate and molybdenum can effectively improve their nitrogen metabolism and carbon fixation.

### 5. Conclusions

During this study, it was demonstrated that nitrate absorption reached its highest level after 56 h of spraying, even at low concentrations. However, nitrate reductase was not activated by sodium molybdate or potassium nitrate application when sprayed for 56 h. Our study demonstrated that the activation of nitrate reductase required the application of molybdate or nitrate for 7 days. The optimum levels of in vivo and in vitro activity of NR were demonstrated within 14 days. This shows that the application of molybdate and nitrite upregulated nitrate reductase activity, increasing nitrate assimilation. Our finding is essential for the growing of plants under aquaponic conditions.

**Author Contributions:** K.A. drafted the manuscript; B.Z., U.S., A.T. and A.B. participated in the analysis of the data; K.A., K.Z.B., A.M., M.K., K.M.A. and A.N. participated in conducting the experiments; M.S. performed the experiments and participated in writing the manuscript; Z.N. participated in writing the manuscript and in editing; Z.A. conceived the original idea, designed the research plan and supervised the research work. All authors have read and agreed to the published version of the manuscript.

**Funding:** This research was funded by the Ministry of Science and Higher Education of the Republic of Kazakhstan (Grant numbers AP09260589 and AP19680579).

**Data Availability Statement:** All the data available within the manuscript.

**Acknowledgments:** We are very grateful to Ralf-Rainer Mendel, University of Braunschweig, for providing the *Neurospora crassa* nit-1 mutant strain.

**Conflicts of Interest:** The authors declare no conflict of interest.

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
