# Peer review of "Foliar Fertilization with Molybdate and Nitrate Up-Regulated Activity of Nitrate Reductase in Lemon Balm Leaves"

_horticulturae, doi:10.3390/horticulturae9121325_

Round 1

Reviewer 1 Report

Comments and Suggestions for Authors

1, Redesigning Fig. 1, it is recommended to delete Fig. 1A-C or Fig. 1D because of their duplicated data.
2, It is incomplete to study only the 1.5 µM molybdate treatment.
3, Indicate the method of multiple comparisons, Duncan or Tukey or some other method.
4, The formatting of in vivo and in vitro in figures and figure captions, as well as in analyses, suggests the uniform use of italics.
5, It is suggested to add the study of NR-related gene expression changes.
6, The discussion needs to be further strengthened.

Author Response

Response to reviewer # 1 comments:

Dear reviewer,

We are grateful for your review of our manuscript entitled “Foliar Fertilization with Molybdate and Nitrate Up-regulated Activity of Nitrate Reductase in Lemon Balm Leaves”. Our paper was greatly improved by your suggestions and corrections. Please find the detailed responses below and the corresponding revisions/corrections with highlighted changes in the re-submitted files.

  1. Redesigning 1, it is recommended to delete Fig. 1A-C or Fig. 1D because of their duplicated data.

Respond to 1.

Fig.1D was deleted as recommended.

  1. It is incomplete to study only the 1.5 µM molybdate treatment.

Respond to 2.

We used 1.5 µM molybdate according to previously reported works where this concentration was demonstrated as effective (Kovács, B.; Puskás-Preszner, A.; Huzsvai, L.; Lévai, L.; Bódi, É. Effect of Molybdenum Treatment on Molybdenum Concentration and Nitrate Reduction in Maize Seedlings. Plant Physiology and Biochemistry 2015, 96, 38–44, doi:10.1016/j.plaphy.2015.07.013.).

  1. Indicate the method of multiple comparisons, Duncan or Tukey or some other method.

Respond to 3.

The Tukey was indicated as a method of multiple comparisons.

  1. The formatting ofin vivo and in vitro in figures and figure captions, as well as in analyses, suggests the uniform use of italics.

Respond to 4.

The “in vivo and in vitro” were checked throughout the paper and corrected as required places with italics.

  1. It is suggested to add the study of NR-related gene expression changes.

Respond to 5.

The activity level is preferred since it is closer to the final product and gene expression may not reflect the activity of multicomponent enzymes such as NR.

  1. The discussion needs to be further strengthened.

Respond to 6.

The discussion has been strengthened by the second reviewer's suggestions.

Reviewer 2 Report

Comments and Suggestions for Authors

Dear Authors,

In this article, the authors investigate the effect of adding different concentrations of nitrate and molybdate to Lemon Balm plants using a spray. The article needs very substantial improvements, with significant flaws that need all to be addressed before potential publication.

Majors:

L20: “ Nitrate  activity”???? I suppose the authors meant to refer to the nitrate reductase activity.

L43: “ and sulfite oxidase (SO) [3].” There is another molybdoenzyme in plants, mARC, and it should be cited. DOI: 10.3390/molecules23123287

L48: “nitrate to nitrite [3,4].” Nitrate reductase also performs another function, the production of nitric oxide. DOI: 10.1093/jxb/erw507

-L50: “AO” It is not mentioned that AO participates in the formation of the phytohormone abscisic acid. Doi: 10.1093/jxb/erv528

L51: “While, SO is the only enzyme in plants that has been thoroughly studied at the molecular and biochemical levels [9,10].” This is not correct; other molybdoenzymes have also been studied in more depth.

L54: “Mo-enzymes catalyze, almost exclusively, oxidation/reduction reactions, and have a  predominantly homo-dimeric structure consisting of two subunits [2–10]. Each of the two  subunits contains two iron-sulfur centers, flavin adenine dinucleotide (FAD) and molybdenum cofactor (Mo-Co) [2–10]” This is not correct; the type of domains varies among different molybdoenzymes. Check it here: DOI 10.1002/biof.1362

L59:” The Mo atom is incorporated into Mo-Co during  the assembly of the Mo-enzymes molecule [11,12]” Those references are not correct; this is where the process is described: DOI: 10.1074/jbc.M601415200

L61: …. I advise the authors to include a figure of the molybdenum cofactor structure to enhance the understanding of this description

L61: “monosaccharide,”??? I don't understand what the authors mean by this.

L64: “Thus, property of all Mo-enzymes is the cofactor content of their active center” Rewrite, it is poorly expressed

L73: “Foliar fertilization is most effective when the availability  of nutrients in the soil growth médium”??? It doesn't make sense, please rewrite

L77: “Another source of nitrogen is ammonium, when an excess amount becomes toxic to  aquatic animals in the aquatic environment”??? I don't understand the why and the meaning of this sentence here.

- The introduction does not mention that there are proteins described that accumulate and protect Mo-Co in plants once synthesized. Doi: 10.3390/molecules27196571

L104: “with 13 h of illumination,”?? but if it has been previously mentioned that the photoperiod is 16 hours.

L142: “in the same buffer which was used to separate proteins from low molecular weight compounds that could affect enzyme activity” I don't understand what the authors mean by this; please clarify.

L146:” buffer A”???

L149: “hypoxanthine”??? why??

L152:” nitrate content” I guess nitrite

-In my understanding, figures F1A, B, and C are redundant since all that information is included in D

-L208: “38 h” I guess 32h

-L215: “Wherein in vivo and in vitro NR activity was not detected in  leaves 56 h after spraying with nitrate at the indicated concentrations” And where are those data, what do they refer to?

L245: “because the cofactor content increases due to new syntheses of Mo-Co by NR molecules.” This sentence doesn't make sense; rewrite it. It seems like they are saying that NR is the one synthesizing Mo-Co.

L251: “AO, XDH, and SO [6–10].” And the mARC

-In the legend of Fig 2: NO2-??? I think they mean nitrate. Please indicate the units of Mo-Co quantity on the graph.

-L266: “hypoxanthine—XDH substrate” I don't see the point, if they quantify NR, why add hypoxanthine when it's the substrate for XDH?

-L267: “animal XDH”??? But what does this have to do with plants?

L290: “The leaf extracts of PF with nitrate did not show such activity, since plant XDH  does not have NR activity, unlike the animal enzyme.”??? I advise the authors to remove everything related to hypoxanthine from the paper; I don't see any sense in it.

-L293:” The use of NR activity of leaves as a marker to establish the incorporation of exogenous molybdenum into the Mo-enzyme molecule.”??? This sentence is not finished; what do they mean?

L297: “Among all Mo-enzymes, 295 NR is inducible, which means that synthesis of NR occurs only in the presence of its sub- 296 strate, nitrate, in the cell [3,4].” This is not correct; NR is also induced in media without ammonium, that is, in nitrogen-free media. DOI 10.3390/plants907090

L305: “in the cofactor domain.” ??? In which of them?

L329: “NR can modulate” I guess, NR can be modulate

L347: “nitrite” I guess nitrate

The references have a tremendous amount of errors. At first glance, I have verified the following: All references must be checked again by the authors one by one, and with more care this time.

-References 34 and 44 are repeated

-References 31 and 36 are repeated

-References 30 and 35 are repeated

-References 32 and 37 are repeated

Please, all references must be checked again by the authors one by one, with more care.

Minors:

L36: “ molecule'’s conformation” tipo

-L80” molybdenum Mo” Redundant

-vivo and in vitro:  Check it throughout the paper; it should be in italics

-L335: “ihts” typo

Author Response

Response to reviewer # 2 comments:

Dear reviewer,

We are grateful for your review of our manuscript entitled “Foliar Fertilization with Molybdate and Nitrate Up-regulated Activity of Nitrate Reductase in Lemon Balm Leaves”. Our paper was greatly improved by your suggestions and corrections. Please find the detailed responses below and the corresponding revisions/corrections with highlighted changes in the re-submitted files.

  1. L20: “Nitrate activity”???? I suppose the authors meant to refer to the nitrate reductase activity.

Respond to 1.

Now, it is written Nitrate reductase activity in lines 20-21.

  1. L43: “and sulfite oxidase (SO) [3].” There is another molybdoenzyme in plants, mARC, and it should be cited. DOI: 10.3390/molecules23123287

Respond to 2.

We cited as suggested. Now, it is written: sulfite oxidase (SO), and mitochondrial Amidoxime Reducing Component (mARC) [3,4].” in line 44.

  1. L48: “nitrate to nitrite [3,4].” Nitrate reductase also performs another function, the production of nitric oxide. DOI: 10.1093/jxb/erw507

Respond to 3.

Now, it is written: “NR catalyzes the reduction of nitrate to nitrite and performs nitric oxide production [3,5,6].” in lines 50-51.

  1. -L50: “AO” It is not mentioned that AO participates in the formation of the phytohormone abscisic acid. Doi: 10.1093/jxb/erv528

Respond to 4.

Now, it is written: “AO oxidizes aldehydes and heterocyclic compounds to corresponding carboxylic acids and catalyzes the oxidation of abscisic aldehyde to abscisic acid [9–11].” in lines 52-54.

  1. L51: “While, SO is the only enzyme in plants that has been thoroughly studied at the molecular and biochemical levels [9,10].” This is not correct; other molybdoenzymes have also been studied in more depth.

Respond to 5.

The following sentence: “While, SO is the only enzyme in plants that has been thoroughly studied at the molecular and biochemical levels [9,10]” was deleted. Now, it is written: “SO oxidizes sulfite with molecular oxygen as an electron acceptor, producing sulfate and hydrogen peroxide [13,14]” in lines 53-55.

According to your advice following sentences were added: “However, the function of the most recently discovered Mo-enzyme mARC is still being investigated [15]. It is now known that its functions include the reduction of hydroxylated compounds, as well as nitrites to nitric oxide [4,15].” in lines 55-57.

  1. L54: “Mo-enzymes catalyze, almost exclusively, oxidation/reduction reactions, and have a  predominantly homo-dimeric structure consisting of two subunits [2–10]. Each of the two  subunits contains two iron-sulfur centers, flavin adenine dinucleotide (FAD) and molybdenum cofactor (Mo-Co) [2–10]” This is not correct; the type of domains varies among different molybdoenzymes. Check it here: DOI 10.1002/biof.1362

Respond to 6.

Now, it is corrected and written as follows:

“Various Mo-enzymes perform vital functions for normal plant development and catalyze, almost exclusively, oxidation/reduction reactions [2–15].” in lines 47-48.

“The plant Mo-enzymes have a predominantly homo-dimeric structure consisting of two subunits (mARC is an exception) [4–15]. However, plant XDH, AO, NR, and SO have differing domains in the subunits depending on the type of Mo-enzymes [5–15]. In XDH and AO enzymes, each of the two subunits contains two iron-sulfur centers, flavin adenine dinucleotide (FAD) and molybdenum cofactor (Mo-Co) [7–12]. The NR contains five conserved domains in each monomer [5,6]. These domains include the Mo-molybdopterin domain, which has a single Мо-Со, a dimer interface domain, a cytochrome b domain, and the NADPH/NADH domain, which combines with the FAD domain to create the cytochrome b reductase fragment [5,6]. SO as the simplest plant Mo-enzyme possesses only one redox center in the form of Mo-Co [13,14]. Mainly monomeric mARC generally in the eukaryotes have two domains: a MOSC domain, which is found in MOS enzymes involved in Moco sulfurases, and a β-barrel domain [4,15].” in lines 58-69.

  1. L59:” The Mo atom is incorporated into Mo-Co during  the assembly of the Mo-enzymes molecule [11,12]” Those references are not correct; this is where the process is described: DOI: 10.1074/jbc.M601415200

Respond to 7.

References are corrected. Now, it is written as follows: “The Mo atom is incorporated into Mo-Co during the assembly of the Mo-enzymes molecule [18,19].” in lines 72-73.

  1. L61: “monosaccharide,”??? I don't understand what the authors mean by this.

Respond to 8.

The “monosaccharaide” was deleted. Now, it is written as follows: “These cofactors are called pyranopterin and have a tricyclic structure comprising pyrimidine, piperazine, and pyran-dithiolene rings (Figure 1) [2–4,16]” in lines 73-75.

  1. L61: …. I advise the authors to include a figure of the molybdenum cofactor structure to enhance the understanding of this description.

Respond to 9.

The figure was added as requested, in the lines 106-109.

      10. L64: “Thus, property of all Mo-enzymes is the cofactor content of their active center” Rewrite, it is poorly expressed.

Respond to 10.

The following sentence: “Thus, property of all Mo-enzymes is the cofactor content of their active center” was deleted. Instead, following sentence: “However, the Mo-Co structures can vary according to Mo-enzyme type. In the case of SO, NR, and mARC, Mo-Co directly binds with apoenzymes, but in the case of AO or XOR/XD, Mo-Co, a final sulfuration undertakes to incorporate into the apoenzymes (Figure 1) [4]” was added in lines 78-81.

      11. L73: “Foliar fertilization is most effective when the availability of nutrients in the soil growth médium”??? It doesn't make sense, please rewrite

Respond to 11.

Now, it is written as follows: “Foliar fertilization is most effective when soil nutrients are deficient [26]” in lines 90-91.

      12. L77: “Another source of nitrogen is ammonium, when an excess amount becomes toxic to  aquatic animals in the aquatic environment”??? I don't understand the why and the meaning of this sentence here.

Respond to 12.

The sentence was deleted.

      13. - The introduction does not mention that there are proteins described that accumulate and protect Mo-Co in plants once synthesized. Doi: 10.3390/molecules27196571

Respond to 13.

The following sentence was added: “Additionally, there are some MoCo binding proteins which has property to interact and transfer MoCo [20].” in lines 81-82.

      14. L104: “with 13 h of illumination,”?? but if it has been previously mentioned that the photoperiod is 16 hours.

Respond to 14.

Corrected as “16 h” in line 123.

      15. L142: “in the same buffer which was used to separate proteins from low molecular weight compounds that could affect enzyme activity” I don't understand what the authors mean by this; please clarify.

Respond to 15.

Now, it is written: “For the determination of NR and Mo-Co activity, leaf extracts were passed through Sephadex G-25 (course, Pharmacia Fine Chemicals) in the extraction buffer. Sephadex G-25 was used to separate proteins from low molecular weight compounds that could affect enzyme activity.” in lines 160-163

      16. L146:” buffer A”???

Respond to 16.

Now, it is written as follows: “To determine NR activity, 200 µL of supernatant was added to 300 µL of PBS pH 7.0 containing 10 μM EDTA and 20 μM FAD to obtain a final volume of 0.5 mL [33,41].” in lines 167-168.

      17. L149: “hypoxanthine”??? why??

Respond to 17.

The “hypoxanthine” was deleted.

      18. L152:” nitrate content” I guess nitrite

Respond to 18.

The “nitrate content” was corrected to “nitrite content” in line 174.

      19. -In my understanding, figures F1A, B, and C are redundant since all that information is included in D

Respond to 19.

We agree with the suggestion of the reviewer. Figure 1D was deleted and legend of the Figure 1 was edited.

      20. -L208: “38 h” I guess 32h

Respond to 20.

“38 h” was corrected to 32 h in line 223.

      21. -L215: “Wherein in vivo and in vitro NR activity was not detected in  leaves 56 h after spraying with nitrate at the indicated concentrations” And where are those data, what do they refer to?

Respond to 21.

Now, it is written as follows: “Wherein in vivo and in vitro NR activity was not detected in leaves 56 h after spraying with nitrate at the indicated concentrations (data not shown).” in lines 230-231.

      22. L245: “because the cofactor content increases due to new syntheses of Mo-Co by NR molecules.” This sentence doesn't make sense; rewrite it. It seems like they are saying that NR is the one synthesizing Mo-Co.

Respond to 22.

Now, it is written as follows: “As shown in Figure 3, from 24 h after nitrate spraying, Mo-Co activity increases because the cofactor content increases. It is due to new syntheses of Mo-Co in NR molecules” in lines 267-269.

      23. L251: “AO, XDH, and SO [6–10].” And the mARC

Respond to 23.

Now, it is written as follows: “The source of Mo-Co in control leaves (without nitrate) comes from the other Mo-enzymes such as AO, XDH, SO, and mARC [2–15]” in lines 272-274.

      24. -In the legend of Fig 2: NO2-??? I think they mean nitrate. Please indicate the units of Mo-Co quantity on the graph.

Respond to 24.

Units of Mo-Co quantity indicated in the graph.

      25. -L266: “hypoxanthine—XDH substrate” I don't see the point, if they quantify NR, why add hypoxanthine when it's the substrate for XDH?

Respond to 25.

The “hypoxanthine—XDH substrate” was deleted.

       26. -L267: “animal XDH” ??? But what does this have to do with plants?

Respond to 26.

The “animal XDH” was deleted as suggested.

      27. L290: “The leaf extracts of PF with nitrate did not show such activity, since plant XDH  does not have NR activity, unlike the animal enzyme.”??? I advise the authors to remove everything related to hypoxanthine from the paper; I don't see any sense in it.

Respond to 27.

Everything related to hypoxanthine was deleted as suggested.

      28. -L293:” The use of NR activity of leaves as a marker to establish the incorporation of exogenous molybdenum into the Mo-enzyme molecule.”??? This sentence is not finished; what do they mean?

Respond to 28.

Now, it is written as follows: “In this work, the NR activity of leaves was used as a marker to establish the incorporation of exogenous molybdenum into the Mo-enzyme molecules” in lines 308-309.

      29. L297: “Among all Mo-enzymes, 295 NR is inducible, which means that synthesis of NR occurs only in the presence of its sub- 296 strate, nitrate, in the cell [3,4].” This is not correct; NR is also induced in media without ammonium, that is, in nitrogen-free media. DOI 10.3390/plants907090

Respond to 29.

Now it is written as follows: “Plant NR is an inducible Mo-enzyme that synthesizes in the presence of its substrate, nitrate, within the cell [3,5,46–48]” in lines 310-312.

The offered paper was useful, and was cited in following sentence: “NR is the first enzyme of the nitrate nitrogen assimilation pathway in plants [2–5,51]” in lines 344-345.

      30. L305: “in the cofactor domain.” ??? In which of them?

Respond to 30.

Now, it is written as follows: “It should be noted that nitrate reduction occurs in the Mo-Co domain” in lines 319-320.

      31. L329: “NR can modulate” I guess, NR can be modulate

Respond to 31.

The “NR can modulate” corrected as “NR can be modulated” in lines 345-346.

      32. L347: “nitrite” I guess nitrate

Respond to 32.

It was corrected as you recommend in line 361.

      33. The references have a tremendous amount of errors. At first glance, I have verified the following: All references must be checked again by the authors one by one, and with more care this time.

Respond to 33.

References checked and corrected one by one by using reference manager software Zotero.

      34. -References 34 and 44 are repeated

Respond to 34.

Repeated reference was corrected.

      35. -References 31 and 36 are repeated

Respond to 35.

Repeated reference was corrected.

       36. -References 30 and 35 are repeated

Respond to 36.

Repeated reference was corrected.

      37. -References 32 and 37 are repeated

Respond to 37.

Repeated reference was corrected.

      38. Please, all references must be checked again by the authors one by one, with more care.

Respond to 38.

References checked and corrected one by one.

      39. L36: “ molecule'’s conformation” tipo

Respond to 39.

Corrected as suggested.

      40. -L80” molybdenum Mo” Redundant

Respond to 40.

Corrected as suggested. Now, it is written only “Mo” in line 95.

      41.-vivo and in vitro:  Check it throughout the paper; it should be in italics

Respond to 41.

The “in vivo and in vitro” were checked throughout the paper and corrected as required places with italics.

      42.-L335: “ihts” typo

Respond to 42.

Corrected as requested.

Round 2

Reviewer 1 Report

Comments and Suggestions for Authors

Authors have addressed all the queries raised during review process.

Reviewer 2 Report

Comments and Suggestions for Authors

Dear Authors,

I believe the authors have correctly addressed all of my changes and suggestions, and I accept the paper in its current version.